# Preparation of MnO_2_-Carbon Materials and Their Applications in Photocatalytic Water Treatment

**DOI:** 10.3390/nano13030541

**Published:** 2023-01-29

**Authors:** Kun Fan, Qing Chen, Jian Zhao, Yue Liu

**Affiliations:** 1Chinese Research Academy of Environment Sciences, Beijing 100012, China; 2Ecological and Environmental Protection Company, China South-to-North Water Diversion Corporation Limited, Beijing 100036, China

**Keywords:** water pollution, MnO_2_-carbon materials, photocatalytic, environmental applications

## Abstract

Water pollution is one of the most important problems in the field of environmental protection in the whole world, and organic pollution is a critical one for wastewater pollution problems. How to solve the problem effectively has triggered a common concern in the area of environmental protection nowadays. Around this problem, scientists have carried out a lot of research; due to the advantages of high efficiency, a lack of secondary pollution, and low cost, photocatalytic technology has attracted more and more attention. In the past, MnO_2_ was seldom used in the field of water pollution treatment due to its easy agglomeration and low catalytic activity at low temperatures. With the development of carbon materials, it was found that the composite of carbon materials and MnO_2_ could overcome the above defects, and the composite had good photocatalytic performance, and the research on the photocatalytic performance of MnO_2_-carbon materials has gradually become a research hotspot in recent years. This review covers recent progress on MnO_2_-carbon materials for photocatalytic water treatment. We focus on the preparation methods of MnO_2_ and different kinds of carbon material composites and the application of composite materials in the removal of phenolic compounds, antibiotics, organic dyes, and heavy metal ions in water. Finally, we present our perspective on the challenges and future research directions of MnO_2_-carbon materials in the field of environmental applications.

## 1. Introduction

With the rapid development of modern industry and agriculture and the rapid growth of population, agricultural, industrial and domestic water use has increased tremendously [1,2]. Refractory toxic pollutants such as pesticides [3], antibiotics [4], textile dyes [5], and heavy metals [6,7] are discharged into water bodies, posing a huge threat to aquatic ecosystems and human health. Water pollution has become among the most pressing issues in the whole world [8]. Hence, green, highly efficient, and low-cost water treatment technologies are in urgent demand. Photocatalysis has been recognized as an ideal tool to eliminate recalcitrant contaminants in aqueous environments owing to its high efficiency, energy savings, low cost, environmental friendliness, lack of secondary pollution, and other characteristics [9,10,11].

Photocatalytic materials are the core of photocatalytic technology [2,12,13,14]. In recent years, semiconductors based on metal oxides are mostly used as photocatalysts for environmental remediation, such as MnO_2_ [15], TiO_2_ [16], ZnO [17], Fe_2_O_3_ [18], SnO_2_ [19], etc. Photocatalytic reactions are initiated by absorbing light energy equal to or more than the bandgap of semiconductor photocatalysts, so the bandgap is an important parameter in defining the applicability of semiconductors in specific photocatalytic reactions [20,21]. Narrow bandgap semiconductors can improve the utilization of visible light, which is more beneficial for water purification applications [22]. Therefore, compared to wide-bandgap semiconductor photocatalysts, narrow-bandgap manganese dioxide (MnO_2_) can degrade organic pollutants under visible light irradiation [23,24]. In addition, MnO_2_ is the most promising environment-friendly photocatalytic candidate material due to its low cost, non-toxic properties, ease of synthesis, rich structures and morphologies, outstanding adsorption, and oxidation capacity [25,26,27]. Cao and Steven [28] first validated its photocatalytic activity through the oxidation of 2-propanol in 1994. However, MnO_2_ has low conductivity, the rate of charge transfer is slow, the photogenerated electron-hole pairs are prone to be recombined, and its efficiency as a photocatalyst is often restricted [29,30]. Meanwhile, the photocatalytic efficiency of MnO_2_ is affected by its crystal form (α-, β-, γ-, δ-, and λ-types), morphology and structure. These factors are directly related to the preparation method, process, and parameters [31,32,33,34,35]. At present, a large number of studies have found that α-MnO_2_ has good photocatalytic performance, and the catalytic efficiency can be further improved after MnO_2_ and carbon are compounded [36,37,38,39,40]. MnO_2_ has good compatibility with carbon materials, so many researchers combine MnO_2_ with carbon materials to improve its photocatalytic efficiency [38,41,42,43].

Carbon-based materials are extensively used in water treatment as they are economical, abundant in nature, and environmentally friendly, and they show many advantages due to their excellent characteristics [44]. Carbon materials have a well-known electron-storage capacity, which can accept photon-excited electrons to promote charge separation and inhibit electron-hole pair recombination [43,45]. As adsorbents, carbon materials can offer a larger surface area and adsorb a large number of pollutants to the catalyst surface [46]. At the same time, as dopants and sensitizers, carbon materials can improve the solar absorbance range of MnO_2_ to improve photocatalytic activity [47]. There is a good coupling effect between carbon materials and MnO_2_, so their composite has become an important field to be explored. Diverse types of MnO_2_-carbon composites have been investigated as photocatalysts to achieve better photocatalytic activity as well as more stable cycling performance. Many researchers have indicated that combining MnO_2_ with carbon-based materials can diminish the recombination of charge carriers and enhance its photocatalytic performance [30,48,49,50].

Graphene [51], graphitic carbon nitride (g-C_3_N_4_) [52], carbon nanotubes (CNT) [53], carbon quantum dots (CQDs) [54], carbon fibers (CFs) [55], and other carbon materials have many unique properties like rich pore structure and active sites, high specific surface area, good electrical conductivity, excellent electron transport and adsorption ability, which are considered as the superior carriers or co-catalyst of semiconductor photocatalysts [56,57,58,59].

In recent years, multi-component composites based on MnO_2_ and carbon materials have become a research hotspot in the application field of water treatment. There are many articles on the synthesis and application research of MnO_2_-carbon materials, but it is a big challenge to choose a suitable preparation process to make it more suitable for specific applications. We studied the photocatalytic degradation of organic compounds by MnO_2_-graphene. The photocatalytic efficiency of MnO_2_-graphene three-dimensional (3D) composites prepared by thermal reduction was as high as 92%. The primary goal of this review is to investigate the current application of MnO_2_-carbon materials for comprehensive adsorption and photocatalytic treatment of water. We summarize the preparation methods of different types of carbon materials combined with MnO_2_, then analyze the application development of MnO_2_-carbon composites in photocatalytic degradation of various refractory organic or inorganic pollutants in water, last, we discuss the existing problems and future prospects.

## 2. Preparation Methods of MnO_2_-Carbon Composites

### 2.1. Hydrothermal Method

The hydrothermal method with water as the reaction medium has become one of the common methods to prepare MnO_2_-carbon composites because of its economic simplicity and environmental protection [60,61,62]. Nanoparticles with different particle sizes, crystal forms, and morphologies can be obtained by adjusting hydrothermal conditions with high reactivity, controllable conditions, and various synthesis types. In addition, the closed environment with high temperature and high pressure can effectively enhance the close contact between MnO_2_ and carbon materials, improve the transmission speed of electrons, and improve the photocatalytic activity of the composite materials to some extent [63,64]. The hydrothermal method is widely used, which can prepare different dimensions and types of MnO_2_-carbon composites [65,66,67,68,69,70,71,72,73,74,75].

For example, Chhabra et al. [43] prepared α-MnO_2_-RGO nanocomposite by a facile hydrothermal method with RGO reduced by chemical reduction (Figure 1a). In the nanocomposite, the one-dimensional (1D) rod-shaped MnO_2_ increases the flow of electrons in the longitudinal direction and reduces the possibility of electron-hole pair bonding. On the other hand, two-dimensional (2D) RGO nanosheets have a large surface area and pore volume, which can prevent charge recombination by aiding in the quick transport of the charges. With the introduction of RGO nanosheets, the surface area of the material increased to 87.159 m^2^g^−1^, and the composite photocatalyst exhibited efficient adsorptive photocatalytic performance. Wang et al. [76] prepared CNT-MnO_2_ composite film by depositing MnO_2_ nanosheets on CNT film using the hydrothermal method. Under different hydrothermal times, the coverage of MnO_2_ on CNTs films will change. The optimized composite film can be folded into different sizes and shapes, exhibiting excellent flexibility and stability. Doping metal or non-metal on carbon materials or MnO_2_ can incorporate the unique characteristics of different materials to improve performance [77,78,79,80,81,82,83,84,85,86]. Shan et al. [87] first prepared K and Na atom doped g-C_3_N_4_ via the thermal treatment of thiourea and KBr/NaBr, respectively, and then added them into KMnO_4_ solution for hydrothermal reaction to synthesize K/Na doped g-C_3_N_4_@MnO_2_ composite. The MnO_2_ nanosheets were vertically assembled on the surface of g-C_3_N_4_ with a stable structure and shortened the diffusion path lengths for electrons. Metal atoms intercalated into the g-C_3_N_4_ interlayers, which enhanced the conductivity, served as the charge transfer channel between adjacent layers to promote charge transfer and hinder the recombination of photogenerated carriers.

In addition to 2D composites, nano-sized MnO_2_ can be uniformly incorporated into the porous structure of 3D carbon materials via the hydrothermal method, thus improving the photocatalytic activity of hybrid catalysts [93,94,95,96,97]. Nui et al. [98] synthesized Graphene/nano α-MnO_2_ hybrid aerogel in an isopropanol-water system via hydrothermal-thermal reduction. The needlelike α-MnO_2_ nanoparticles are covalently bonded with graphene without damaging the integrity of the graphene structure and are doped in the graphene aerogel uniformly. Due to the porous structure of the hybrid aerogel and the high dispersibility of the MnO_2_ on graphene, the as-prepared composite exhibits good catalytic activity. Wan et al. [88] prepared flower-like core-shell MnO_2_-coated carbon aerogels via the hydrothermal method as a superior photocatalyst to remove organic dyes from an aqueous solution (Figure 1b). Dong et al. [99] prepared 3D MnO_2_/N-doped graphene hybrid aerogel by self-assembly. MnO_2_ nanosheets and nanotubes were first synthesized by a double aging method and hydrothermal method, respectively, then N-doped graphene aerogels were created via hydrothermal-freeze drying process using ethylenediamine as the reductant and nitrogen source. The size and morphology of MnO_2_ play an important role in tailoring the structures and properties of 3D graphene aerogels. The laminar structure of MnO_2_ nanosheets with the graphene conductive substrate is beneficial to enhancing the charge transfer, shortening the diffusion pathway of pollutants, and affording more active sites. However, excessive MnO_2_ nanosheets on graphene might aggregate and inactivate, which adversely affects the overall catalytic activity.

In order to further improve the properties of MnO_2_-carbon materials, many researchers also add green and economical polymers, oxides, or other carbon materials to the MnO_2_-carbon materials to prepare ternary composites, and the hydrothermal method is the most common preparation scheme [100,101,102]. For example, Iqbal et al. [103] prepared PANI@CNT/MnO_2_ ternary composite with rough interwoven fibrous and porous structure by the combination of hydrothermal methodology and in situ oxidative polymerization of aniline. The synergistic effect of the three enhances the specific surface area, thermal and electrical conductivity, and provides channels for the transport of charge carriers, thus enhancing the performance of the material. Wang et al. [89] also utilized the α-Fe_2_O_3_ core/shell configuration to modify g-C_3_N_4_, and prepared a dual Z-scheme α-Fe_2_O_3_@MnO_2_/g-C_3_N_4_ ternary composite by two-step hydrothermal method (Figure 1c). The Fe_2_O_3_@MnO_2_ core/shell promoter modulates the electronic structure through the dual Z-scheme heterojunction, thus improving the separation efficiency of photo-generated electron-hole pairs. Due to its narrow bandgap, the composite material has a broad absorption in the visible light region, low cost and excellent performance, which is more conducive to practical application. Xu et al. [104] selected CC as the substrate for the growth of MnO_2_, coated RGO on the surface of CC, and synthesized CC/RGO/MnO_2_ composites by dipping method and hydrothermal method. CC/RGO can provide a large specific surface area as the skeleton, and the good conductivity of carbon materials can accelerate electron transfer, the resulting composite shows good photoelectrochemical activity. Li et al. [105] synthesized CNT/rGO@MnO_2_ particles through a hydrothermal reaction and then obtained a sandwich-like film with a 3D multilevel porous conductive structure via vacuum filtration and freeze-drying treatment. Nano-sized pores increase the specific surface area and provide a large number of active sites. MnO_2_ grows in situ on the carbon skeleton, and the two are tightly connected, which facilitates electron transportation and enhances structural stability.

Solvothermal and microwave irradiation are improved methods of hydrothermal synthesis [61]. Among them, the solvothermal method is based on the same principle as the hydrothermal method. As water-sensitive compounds cannot be synthesized by the hydrothermal method, water can be replaced with an organic solvent to carry out the reaction [106,107,108,109]. For example, Asif et al. [90] synthesized an urchin-like morphology Ni-doped MnO_2_/CNT nanocomposites by a one-step solvothermal reaction (Figure 1d). Ni doping enhances the conductivity of MnO_2_ and increases the surface area and cycle stability of the composite. Microwave irradiation heating reduces energy consumption compared to hydrothermal reactions, effectively shortens the synthesis time of complexes, and improves product homogeneity [110,111,112]. For example, Sivaraj et al. [110] reported a microwave-assisted process to synthesize the hybrid CNTs-MnO_2_ nanocomposite. The dispersed MnO_2_ nanospheres are uniformly attached to the CNTs' side walls, and a synergistic effect increases the light absorption range, promotes charge separation, and enhances stability.

### 2.2. In Situ Redox Deposition

Although hydrothermal self-assembly is economical and environmentally friendly, and widely used, it requires high temperature and pressure and a long reaction time, which is not suitable for large-scale production and applications [113]. In situ redox deposition is mild, simple, and suitable for the compounding of a wide range of metal oxides with carbon materials, which is another important method for the preparation of MnO_2_-carbon composites. It uses the carbon material as a substrate and involves the in situ deposition of MnO_2_ nanostructures onto the surface of carbon materials through a redox reaction to form a nanocomposite, using the carbon material as a substrate, where the MnO_2_ has uniformly and tightly adhered to the surface of the carbon material [114,115,116,117,118,119,120,121,122,123,124,125].

For example, Qu et al. [91] adopted the modified Hummers method and prepared a pristine GO/MnSO_4_ suspension, then the pristine suspension of GO/MnSO_4_ was in situ transformed into GO/MnO_2_ composites in combination with KMnO_4_, and finally further into RGO/MnO_2_ composites by means of glucose-reduction (Figure 1e). Singu et al. [126] synthesized CNTs-MnO_2_ nanocomposites through the in situ reduction of KMnO_4_ using MWCNTs as the reducing agent and supporting substrate. During the preparation process, the loading of MnO_2_ can be adjusted by varying the amount of KMnO_4_, thereby optimizing the performance of the composite material. Wang et al. [127] adsorbed Mn^2+^ on the surface of g-C_3_N_4_ through the NH_2_ groups in g-C_3_N_4_ for the first time, underwent a redox reaction with KMnO_4_, and synthesized a novel 2D MnO_2_/g-C_3_N_4_ heterojunction composite by in situ deposition of δ-MnO_2_. The bandgaps of MnO_2_ and g-C_3_N_4_ synthesized have a wide visible light response and light absorption range, which are 1.56 eV and 2.69 eV, respectively. At the same time, the matched band structures and the heterojunctions with solid (C-O) bonding between them interface promoted the transfer/separation of photogenerated charge carriers, enhanced the light-harvesting ability, thus the photocatalytic activity can be greatly enhanced. Peng et al. [128] also synthesized N-doped CNT (NCNT) by chemical vapor deposition, and deposited MnO_2_ onto the NCNT surface using in situ oxidation to prepare MnO_2_/NCNT composites. The synergistic effect of MnO_2_ and NCNT obviously improved interfacial electron transfer, which can replace noble metals for the catalytic oxidation of organics.

In carbon materials, the existence of π-π interactions and van der Waals forces between the graphene nanosheets make it easy to aggregate and stack during processing, resulting in reduced surface areas and hidden active sites [129,130]. The quantitative loss of nanoscale materials during the recycling process may influence the fate of adsorbed contaminants, thus causing potential environmental risks [131]. The porous structure of composite films and aerogels can prevent the aggregation of nanosheets and afford more active sites for pollutant diffusion and oxidation. The structure is stable and easily recycled for reuse, which is a superior support for MnO_2_ in the treatment of water, while in situ deposition of MnO_2_ can also improve the mechanical and electron transport properties of carbon materials [132,133,134,135,136,137]. For example, Lv et al. [92] used 3D CQDs/graphene composite aerogels formed by the hydrothermal method as the reducing agent, which reacted with KMnO_4_ to synthesize stable MnO_2_/CQDs/graphene composite aerogel (Figure 1f). The 3D network structure avoided the reunion of the graphene nanosheets and the MnO_2_ nanoparticles, and the CQDs served as a bridge for connecting MnO_2_ and graphene, which effectively improved the conductivity and stability of the composite. Jyothibasu et al. [138] prepared cellulose/f-CNT/MnO_2_ composite films via the direct redox deposition method to uniformly grow MnO_2_ nanostructures on cellulose/functionalized CNT (f-CNT) conductive substrates. The synthetic procedure is simple, inexpensive, environmentally friendly, and can be synthesized in large-scale batches. The synthesized materials have unique porous structures, large specific surface areas, and excellent conductivities.

### 2.3. Electrochemical Deposition

Electrochemical deposition is an effective strategy for the synthesis of nanoscale materials and functionalized composites [106] and has been widely used in synthesizing carbon materials such as MnO_2_-modified carbon cloth and graphene [139,140,141,142,143,144,145,146]. Zhang et al. [141] synthesized hierarchical MnO_2_ nanostructures on activated carbon cloth via a high-voltage anodic electro-deposition process, and the activated carbon cloth substrate enhanced the conductivity and hydrophilicity of the material. Zhu et al. [147] synthesized PANI@γ-MnO_2_/CC ternary hybrid material via hydrothermal and in situ electrochemical polymerization (Figure 2a). The coating PANI layer with a 3D hierarchical structure provides a high specific surface area (96.3389 m^2^g^−1^), which is higher than that PANI@γ-MnO_2_ (41.8632 m^2^g^−1^) and CC(21.1902 m^2^g^−1^) and accelerates the ion diffusion and electron transfer. Li et al. [148] synthesized mesoporous MnO_2_ with high density pores on carbon aerogels substrate by electrochemical deposition. Mesoporous materials can increase the active sites and enhance the electric conductivity, which is more conducive to the transport of electrons and ions. In photocatalytic applications, they can effectively prevent the recombination of photogenerated electrons and holes to improve photocatalytic activity. At the same time, the obtained MnO_2_/carbon aerogel composites are green, low-cost and good in cycle stability, which have potential research and application value.

### 2.4. Co-Precipitating Method

The chemical co-precipitation method is a simple process, with low calcination temperatures and good homogeneity of the prepared complexes, and is one of the common methods for the preparation of carbon composites at low temperatures [64,152,153,154]. Zeng et al. [155] synthesized 1D α-MnO_2_ nanowires and 2D GO nanowires to prepare α-MnO_2_/GO nanohybrids by mechanical grinding and co-precipitating method. The sub-micron GO sheets can occupy the interspace of the interconnected network of α-MnO_2_ nanowires so that the two can be better combined. By comparing the materials prepared by the two methods, it can be found that the co-precipitating method is more conducive to the tight binding of MnO_2_ and GO and facilitates heat and electron transfer between these two materials. However, mechanical grinding may destroy the layered structure of GO and produce more defects, which is not conducive to photon absorption and electron transfer. Liu et al. [149] first synthesized α-MnO_2_ nanofibres/carbon nanotubes hierarchically assembled microspheres (α-MnO_2_/CNT HMs) via a facile chemical precipitation/spray-granulation combined methodology (Figure 2b). The α-MnO_2_ NFs were homogeneously anchored on a highly conductive CNTs framework, forming a close-packed network structure, which remarkably improved the electron-transfer capability. The composite material has excellent stability and cycling durability, low cost, and wide application prospects. Kumar et al. [156] prepared an Ag-doped MnO_2_-CNT nanocomposite using a co-precipitation route. The spheroidal-shaped Ag nanoparticles covered the CNT surface, and its high surface area to volume ratio provides a large number of active sites, showing excellent adsorption performance. Xia et al. [157] grew MnO_2_ nanosheets in situ on the surface of exfoliated g-C_3_N_4_ nanosheets by a wet-chemical method, forming a 2D/2D g-C_3_N_4_/MnO_2_ heterojunction. The photoinduced electrons in MnO_2_ can combine with the holes in g-C_3_N_4_ to enhance the extraction and utilization of photo-generated carriers and improve the degradation rate of pollutants.

### 2.5. Template Method

The template method is mostly used for the preparation of 3D composites [158,159]. Le et al. [160] used diatomite as a template for the massive production of 3D porous graphene by the chemical vapor deposition method. After removing the template, the 3D graphene was N-doped by a hydrothermal reaction, and then the N-doped 3D porous graphene@MnO_2_ hybrid structure was obtained by deposition of MnO_2_ nanosheets. The MnO_2_ nanosheets with a brushy structure were uniformly deposited on the surface of porous graphene, and the synergistic interactions between them enhanced the stability of the composite. After removing the diatomite, the composite retained the 3D structure and surface features of the diatomite template. Moreover, the abundant edges and defects formed during the template removal process and defects caused by nitrogen doping improve the conductivity and charge transfer rate of the composite. The MnO_2_ nanosheets with a brushy structure were uniformly deposited on the surface of porous graphene, and the synergistic interactions between them enhanced the stability of the composite. Wang et al. [150] fabricated 3D CNT@NCNT@ MnO_2_ composites with unique tube-in-tube nanostructures through the sacrificial template method (Figure 2c). The composite has an N-doped 3D double-carbon layers hollow structure and attaches tightly with MnO_2_ nanoflowers grown on its surface, exhibiting large pores, high conductivity, large specific surface areas, and fast diffusion of electrons. Shan et al. [161] also prepared C-doped g-C_3_N_4_ (CCN) using polyporous melamine foam (MRF) as a template and then exploited the synergistic advantages of 2D architectures, coupled CCN with MnO_2_ nanosheets by a hydrothermal method to prepare efficient CCN@MnO_2_ composite. The doping of carbon promoted electron transfer, and the MRF template can prevent the aggregation of sulfourea crystals, thereby reducing the thickness of CCN nanosheets and increasing the specific surface area (40.2 m^2^g^−1^).

### 2.6. Ultrasonic-Assisted and Sonochemical Methods

The Sonochemical-assisted uses sound energy to agitate the composite solution, causing it to undergo a physical or chemical transformation [162]. It can prevent material stacking, enlarge the interlayer spacing of carbon materials such as graphene, facilitate uniform loading of MnO_2_ and enhance the photocatalytic properties of the synthesized semiconductors [64,106]. As a result, it is widely used to prepare various MnO_2_-carbon nanocomposites [163,164,165,166,167,168,169]. Chai et al. [170] synthesized S,O co-doped graphite, carbonitride quantum dots (S, O-CNQDs) by a solid-state reaction method, and in situ synthesized MnO_2_ nanosheets in S,O-CNQDs dispersion solution to prepare MnO_2_ -S,O-CNQDs nanocomposite with the ultrasonic-assisted. The as-prepared composite material has uniform size and good dispersion, which is a promising nanomaterial. Xu et al. [151] synthesized the CQDs/MnO_2_ nanoflowers through the sonochemical method (Figure 2d), which has a high specific surface area (168.8 m^2^g^−1^) and excellent cycle stability. CQDs were uniformly distributed on the transparent petals of δ-MnO_2_, which improved the conductivity of MnO_2_ nanoflowers and provided a large number of functional groups and active sites.

### 2.7. Other Methods

Many other novel options are also used to prepare MnO_2_-carbon materials [171,172,173]. For example, Jia et al. [174] prepared CNTs/MnO_2_ composites by in situ synthesis of CNTs on MnO_2_ nanosheets using the hydrothermal method and the chemical vapor deposition method. The vertically aligned MnO_2_ nanosheets shortened the ion diffusion path, the in situ formed CNTs improved the electrical conductivity and structural stability, and the hierarchical porous structure increased the specific surface area (20.4 m^2^g^−1^ to 38.2 m^2^g^−1^) and active sites. Abdullah et al. [175] used polyacrylonitrile (PAN) as a carbon precursor to prepare nanofibers (NFs) by an electrospinning process and incorporated MnO_2_ nanoparticles into ACNFs to prepare composite activated carbon nanofibers (ACNFs/MnO_2_) by carbonization and activation. The incorporation of MnO_2_ increased the specific surface area (478.2 to 599.4 m^2^g^−1^), pore size (0.285 cm^3^g^−1^), and total pore volume (0.299 cm^3^g^−1^) of the composite material. Wei et al. [176] prepared MnO_2_/3D graphene composites by the reverse microemulsion method. In this reaction, the graphene substrate was used as a sacrificial reductant to undergo a redox reaction with KMnO_4_ to grow MnO_2_ in situ on 3D graphene, and the MnO_2_ mass loading of the composite was controlled by changing the ultrasonication time in the in situ growth process. Nanoscale MnO_2_ layers were uniformly coated on the internal surface of 3D graphene, and the continuous 3D interpenetrating microstructures prevented the restacking of graphene sheets. Zhu et al. [177] prepared free-standing 3D graphene/MnO_2_ hybrids by depositing MnO_2_ nanosheets onto a 3D graphene framework through a solution-phase assembly process. Unlike 1D MnO_2_, the flower-like architecture of deposited MnO_2_ nanosheets have a larger specific surface area and are uniformly anchored on a 3D graphene framework with strong adhesion, there is a strong interaction between them, so the prepared hybrids showed good mechanical properties. Pang et al. [178] proposed a simple room-temperature water bath method to deposit crystalline MnO_2_ on CNTs to prepare CNT-MnO_2_ nanocomposites. This scheme can control the phases and morphologies of the composite products by changing the pH of the reaction solution. Wang et al. [179] assembled GO, MnOx, and polymer carbon nitride (CN) into free-standing GO/MnO_x_/CN ternary composite film by employing the vacuum filtration method. The prepared composite film has good stability, mechanical property, and recyclability and is more suitable for the practical application of photocatalysis.

To sum up, MnO_2_-carbon composites can be prepared and modified in various ways, and the finally obtained multifunctional materials have great application potential in water treatment. Each preparation method has its own unique advantages, and the fabrication of specific nanocomposites can be improved by selecting the most suitable preparation method, which can be used to treat various types of sewage treatment to different pollutants (Table 1).

## 3. Applications of Photocatalytic Technology in Water Treatment

With the growth of population and continuous development of industry and agriculture, the problem of water pollution has become increasingly prominent [180,181,182]. Various organic and inorganic pollutants have been detected in surface, ground, sewage, and drinking waters [183,184]. Among them, the pollutants (phenols, antibiotics, organic dyes, heavy metal, etc.) produced by agriculture [185], aquaculture [186,187], carbon aerogels, textiles [188,189] and other industries are highly toxic and difficult to biodegrade [180]. The pollution of water bodies will not only destroy the ecosystem but also seriously threaten human health [190,191]. These stubborn compounds have become important contaminants in water that need to be removed urgently. As common green materials, MnO_2_ and carbon materials can use solar energy to degrade many types of pollutants, and the photocatalytic process is economical and environmentally friendly [192]. Therefore, the combination of the two has momentous research potential and application prospects in the field of photocatalytic water treatment, and the photocatalytic degradation of various pollutants by MnO_2_-carbon materials has also been widely studied [193].

### 3.1. Phenolic Wastewater

Phenolic compounds are typical aromatic organic compounds that exist in sewage discharged from petroleum refineries, manufacturing of paints, pulp and paper manufacturing plants, and other industries [194,195,196]. At the same time, they are also a kind of important organic raw materials in the field of agricultural production and are widely used in the manufacture of pesticides, insecticides, and herbicides [197,198,199]. Their wide use in industry and agriculture makes them a large number of residues in the environment and a common organic pollutant in water, which have potential carcinogenicity, teratogenicity, and mutagenicity, with wide source, great harm and refractory degradation [200,201,202]. Among them, phenol and its derivatives (such as bisphenol A, chlorophenol, nitrophenol, etc.) are common phenolic pollutants in the water environment, which are highly toxic and cause serious pollution even at low concentrations [194,203,204,205,206]. Compared with other organic substances, they have a great impact on the environment.

Phenolic compounds usually have one or more hydroxyl groups attached to the aromatic ring [194]. In the photocatalytic process, hydroxyl radicals attack the cyclic carbon to produce various oxidation intermediates (such as hydroquinone, catechol, p-benzoquinone, etc.) [195]. These organic compounds are less harmful than the parent compounds and will eventually be photomineralized to carbon dioxide (CO_2_), so as to achieve the purpose of degradation [207]. Table 2 summarizes the progress in the photocatalytic degradation of phenolic compounds by various MnO_2_-carbon materials. For example, Mehta et al. [207] prepared MnO_2_@CQDs nanocomposites with a bandgap of 1.3 eV by a one-step hydrothermal method, which was used to degrade phenol under visible light. The spherical CQDs were deposited on the surface of MnO_2_ nanorods, and the nanocomposite had a high specific surface area (95.3 m^2^g^−1^). The optimal operating parameters were obtained after optimization under different reaction conditions, and after 50 min of visible light irradiation, the degradation rate of phenol reached 90%. The degradation rate was basically unchanged after three consecutive cycles, and the degradation rate can still reach 80% after five cycles, which the stability is good. Xia et al. [157] synthesized g-C_3_N_4_/MnO_2_ heterostructured photocatalyst via in situ growth of MnO_2_ nanosheets on the surface of exfoliated g-C_3_N_4_ nanosheets using a wet-chemical method. MnO_2_ nanosheets and the g-C_3_N_4_ layers are closely combined, and the 2D layered structure can provide abundant active sites and shorten the transport distance of photogenerated charge carriers. Under the irradiation of xenon lamps, the ability of the composite material to degrade phenol is significantly increased, and it has good durability. Preparing the photocatalyst of the MnO_2_-carbon composites with low bandgap can make full use of solar energy and provide a sustainable green approach for photocatalytic water treatment, and its application potential needs to be further developed [208,209,210,211,212].

### 3.2. Antibiotic Wastewater

Antibiotics can prevent and treat a variety of bacterial infections in humans and animals and are widely used for human beings, animal husbandry, and aquaculture industries [213,214,215]. However, the overuse of antibiotics has imposed severe water environment problems [215]. According to statistics, approximately 60–90% of antibiotics cannot be completely metabolized by the human or animal body and will be excreted through feces [216,217,218,219]. These wastes may be dumped directly into wastewater or enter farmland as fertilizer and enter nearby water bodies through rainfall and irrigation [220,221]. Due to the poor biodegradability of most antibiotics, the sustained use of antibiotics makes them stay in the water for a long time, which may generate antibiotic-resistant genes (ARGs) and antibiotic-resistant bacteria (ARBs), resulting in increased microbial resistance, which poses a potential threat to human health and ecological systems [222,223,224]. Therefore, the degradation of antibiotics in water is an important and urgent task.

It has been reported that sunlight-driven photocatalytic technology can effectively remove antibiotics from water, among which the visible light-responsive MnO_2_-carbon composite photocatalyst has great practical application potential [225,226]. We selected and listed the photocatalytic degradation rates of different types of antibiotics using MnO_2_-carbon as a photocatalyst (Table 3). Du et al. [227] synthesized g-C_3_N_4_/MnO_2_/GO heterojunction photocatalyst by wet-chemical method (Figure 3a). Composites with different ratios of g-C_3_N_4_, MnO_2,_ and GO have different catalytic activities, and the composites, after optimization, can degrade 91.4% of TC at most after 60 min of visible light irradiation. The TC removal rate only decreases by 10% after four cycles, and the sample structure has no change (Figure 3b–d). Excellent stability is more conducive to the practical application of photocatalysis. Liu et al. [228] synthesized the pumice-supported reduced graphene oxide and MnO_2_ (PS@rGO@MnO_2_) as a solid photocatalyst by a two-step hydrothermal method, which can effectively degrade 80% ciprofloxacin within 6 h under simulated sunlight, and the performance was not obviously decreased after three cycles, and all characteristic peaks remained intact, which proved its excellent reusability. In addition, the catalytic performance of PS@rGO@MnO_2_ solid photocatalyst under actual sunlight is comparable to that under simulated sunlight, it has good removal performance for ciprofloxacin in actual natural water, and it can also degrade other antibiotics in water, which has great potential in the treatment of drinking water and surface water.

### 3.3. Dye Wastewater

Dyes can impart or alter the color of a substance, which are widely used in a wide variety of industries, including textile, printing, leather, agriculture, pharmaceutical, and food industries [229,230,231,232]. According to statistics, more than 7 × 10^5^ tons of dyes are produced annually worldwide, and about 15% of dyes will enter the environment with the loss of wastewater during manufacturing and application processes [233,234]. The dyes have a complex structure, high biological toxicity, and are easily soluble in water but have poor biodegradability, which may accumulate in the water environment [132,235]. The colored dyes in the water will affect the transparency of water bodies, absorb and reflect sunlight entering the water, hinder the photosynthesis of aquatic plants and abolish the ecological balance of the water body [230,236]. In addition, its potential carcinogenicity, teratogenicity, and mutagenicity will also cause negative effects on human health [237,238].

Photocatalytic technology has the remarkable ability to degrade and decolorize organic dyes. In environment-cleaning applications, different kinds of semiconductor compounds play an important role in the photocatalytic removal of dyes [229]. In recent years, MnO_2_-carbon materials have shown excellent performances in the photocatalytic degradation of organic dyes, which has attracted extensive research by researchers [239,240,241,242,243,244,245,246,247,248,249] (Table 4). Park et al. [250] synthesized PANI-rGO-MnO_2_ ternary composites by polymerizing aniline with rGO and incorporating MnO_2_. PANI can act as an excellent electron donor and hole conductor, as channels for electron transport and storage, and is a suitable substrate for visible light-responsive photocatalysts. The ternary heterostructure reduced the recombination of photogenerated electron-hole pairs and extended the light absorption range. The composites showed excellent photocatalytic activity, and 90% of methylene blue (MB) could be degraded under visible light irradiation within 2 h (Figure 4). Panimalar et al. [251] constructed the MnO_2_/g-C_3_N_4_ heterostructure, which showed higher photocatalytic activity than pristine MnO_2_ and g-C_3_N_4_ after 100 min of visible light irradiation. The degradation rate of MO could reach 92%. After five cycles, the composite photocatalyst was not obviously inactivated, showing high stability. This sort of material could be used as a photocatalytic practical device for wastewater treatment.

### 3.4. Heavy Metal Wastewater

The high solubility, bioaccumulation, and non-biodegradability of heavy metals make them easily accumulate in living beings through the food chain and drinking water [252,253,254]. The heavy metals entering the organism are easy to bind with essential cellular components such as proteins, nucleic acids, and enzymes, destroying organic cells in the body and endangering the health of organisms and human bodies [195,255]. However, the toxicity, mutagenicity, and carcinogenicity of heavy metals are strongly dependent on the oxidation state [256]. Reducing a high-valence state and highly toxic heavy metal ion into a low-valence state and low-toxic or non-toxic heavy metal ion is an effective way to mitigate the potential hazards of heavy metals [257,258,259].

It has been reported that MnO_2_-carbon materials can be used as photocatalysts to reduce toxic heavy metals to non-toxic metals using light energy. Padhi et al. [260] reported a highly efficient hydrothermal method to fabricate an RGO/α-MnO_2_ nanorod composite, which showed outstanding photoreduction ability. A 97% reduction in Cr(VI) under visible light irradiation for 2 h and no significant loss of photoreduction ability up to the third cycle. Wang et al. [261] prepared MnO_2_@g-C_3_N_4_ composite photocatalyst by compounding MnO_2_ on g-C_3_N_4_ via the hydrothermal method for the treatment of uranium-containing wastewater. Under optimal conditions, the photocatalytic reduction rate of U(VI) reached 96.3% under visible light irradiation for 120 min. There is little research on the photoreduction of heavy metals by MnO_2_-carbon materials, and related applications still need further exploration.

## 4. Conclusions and Outlook

Photocatalytic technology has attracted extensive attention from researchers because of its green, energy-saving, and high efficiency. It is significant to develop low-cost and non-toxic, environmentally friendly photocatalysts. MnO_2_ and carbon materials are commonly green and low-cost materials, the composite methods are simple and diverse, and different methods can synthesize photocatalytic materials of various dimensions and sizes. Compared with a single photocatalyst, the photocatalytic activity of MnO_2_-carbon composites is significantly improved, and a variety of pollutants can be removed efficiently. At present, many synthetic methods have been developed to prepare MnO_2_-carbon materials for degrading various pollutants, but the practical application is still in the early stage, and no major breakthrough has been made. The transition from the laboratory to the actual water body is still facing great challenges. In future research, the following aspects need further exploration and development.

(1) The performance optimization of MnO_2_-carbon materials. The photocatalytic efficiency of MnO_2_-carbon composites is mostly around 80% or 90%, and the photocatalytic activity needs to be improved further. Therefore, the improvement of photocatalytic performance of MnO_2_-carbon materials is the core problem of photocatalytic technology improvement, and the proportion and preparation process of MnO_2_-carbon composite material fundamentally determine its photocatalytic performance. On the one hand, the properties of MnO_2_-carbon materials can be optimized by adjusting the ratio of MnO_2_ and carbon materials. On the other hand, it can be improved by doping metal or nonmetal, adding polymers, oxides, or other carbon materials.

(2) The separation and recovery of MnO_2_-carbon materials. Powdered MnO_2_-carbon materials not only have the disadvantages that cannot be dispersed evenly and recovered difficulty, but the quantitative loss during the recycling process may influence the fate of adsorbed contaminants, thus causing potential environmental risks. Therefore, it is necessary to explore effective methods to prepare high-dimensional materials that are more conducive to recycling, such as hydrogels, aerogels, and flexible films. Compared with low-dimensional materials, high-dimensional materials have broader prospects in practical applications.

(3) The large-scale application of photocatalytic technology. The application of photocatalytic treatment of MnO_2_-carbon materials mostly stays in the laboratory stage, and it is difficult to use it on a large scale. To achieve large-scale utilization, we need to consider the cost, stability, and quantifiable productivity of photocatalysts. Therefore, it is necessary to explore 3D MnO_2_-carbon materials with better stability, enlarge the size of materials in equal proportion and test their properties, improve the reuse rate of the materials, and reduce the material costs. The stability of MnO_2_-carbon materials and the amplified photocatalytic performance are crucial issues to be solved to realize the large-scale application of photocatalytic technology.

(4) The research on MnO_2_-carbon materials in actual water treatment. Most of the MnO_2_-carbon materials are studied for single pollutants, but the pollutants in actual water bodies have complex components, various kinds, and different concentrations, which are far more complicated than the laboratory simulation. Therefore, we need to evaluate the ability of MnO_2_-carbon materials as a photocatalyst to treat multiple pollutants simultaneously, explore the potential adverse effects of multiple pollutants, develop different sizes and types of MnO_2_-carbon materials, select the study area, collect wastewater samples from actual water bodies, and study the photocatalytic performance of MnO_2_-carbon materials for actual wastewater treatment. The use of MnO_2_-carbon materials for photocatalytic degradation of various organic pollutants in water bodies, from laboratory study to practical water application, is a major challenge and a key research direction for the future.

## Figures and Tables

**Figure 1 nanomaterials-13-00541-f001:**
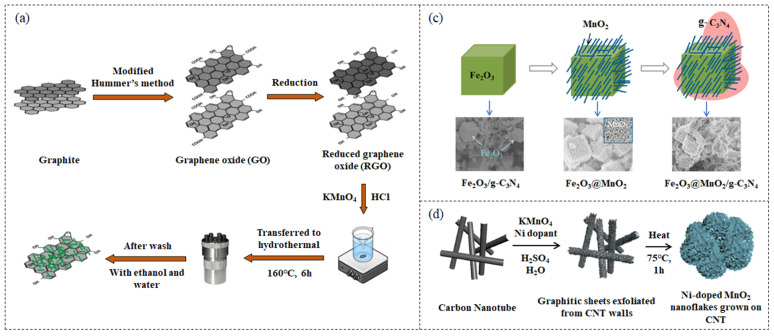
(**a**) Schematic illustration of the preparation of MnO_2_-RGO nanocomposite. Figures reprinted with permission from ref. [43]. Copyright 2019; Elsevier Ltd. (**b**) Schematic diagram of the procedure used to prepare carbon spheres@MnO_2_. Figures reprinted with permission from ref. [88]. Copyright 2019; Elsevier Ltd. (**c**) Schematic illustration of the formation of Fe_2_O_3_@MnO_2_/g-C_3_N_4_. Figures reprinted with permission from ref. [89]. Copyright 2020; Elsevier Ltd. (**d**) Schematic illustration of the growth mechanism of Ni-doped MnO_2_ on CNT. Figures reprinted with permission from ref. [90]. Copyright 2018; Elsevier Ltd. (**e**) Schematic of the synthesis of RGO/MnO_2_ hybrids. Figures reprinted with permission from ref. [91]. Copyright 2014; Elsevier Ltd. (**f**) the fabrication procedure of the MnO_2_/CQDs/graphene composite aerogel. Figures reprinted with permission from ref. [92]. Copyright 2018; Elsevier Ltd.

**Figure 2 nanomaterials-13-00541-f002:**
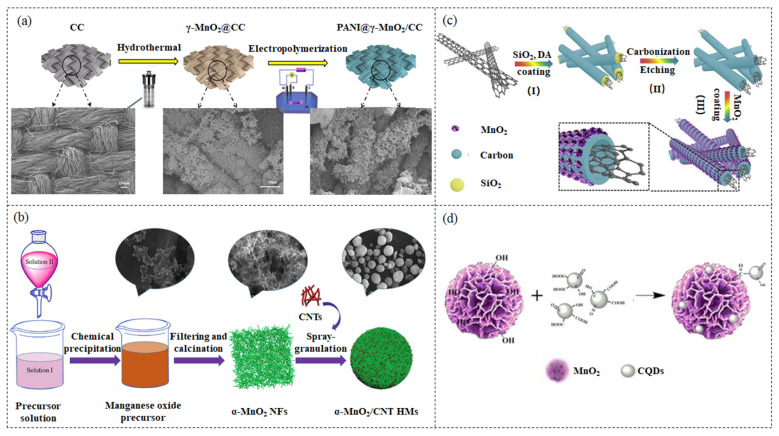
(**a**) The schematic diagram of the process of preparing ternary composite PANI@γ-MnO_2_/CC composite materials and SEM images of different materials. Figures reprinted with permission from ref. [147]. Copyright 2022; Elsevier Ltd. (**b**) Schematic illustration of the preparation of α-MnO_2_/CNT HMs and SEM images at different synthetic stages. Figures reprinted with permission from ref. [149]. Copyright 2019; Elsevier Ltd. (**c**) Schematic illustration of the fabrication of CNT@NCT@MnO_2_ composites. (I) CNTs were sequentially coated with a thick SiO_2_ layer and carbon layer; (II) the removal of the SiO_2_ layer; (III) the growth of ultrathin MnO_2_ nanoflowers on the carbon layer. Figures reprinted with permission from ref. [150]. Copyright 2019; Elsevier Ltd. (**d**) Schematic representation of the preparation of CQDs/MnO_2_ nanoflowers. Figures reprinted with permission from ref. [151]. Copyright 2017; Electrochemical Society.

**Figure 3 nanomaterials-13-00541-f003:**
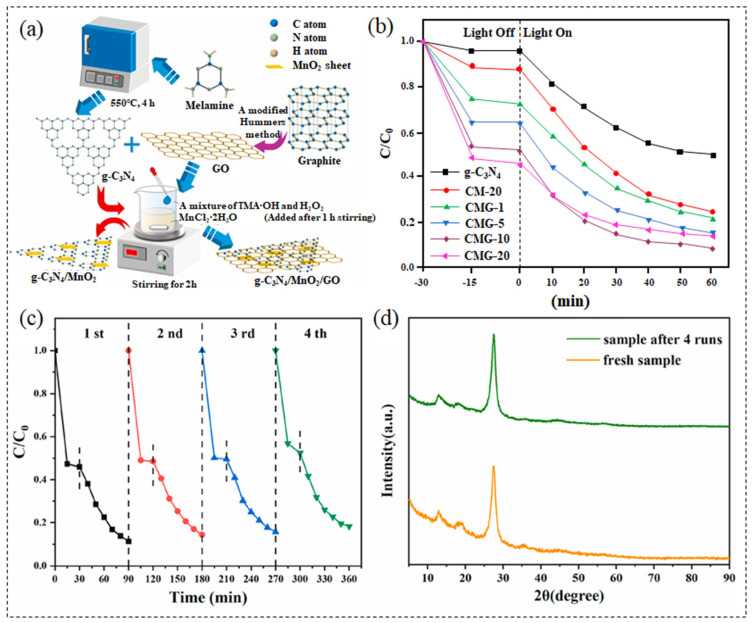
(**a**) Preparation process scheme; (**b**) Photocatalytic degradation rate under the visible light irradiation; (**c**) Recycle experiments for the degradation of TC and (**d**) XRD patterns before and after four runs of CMG-10. Figures reprinted with permission from ref. [227]. Copyright 2021; Elsevier Ltd.

**Figure 4 nanomaterials-13-00541-f004:**
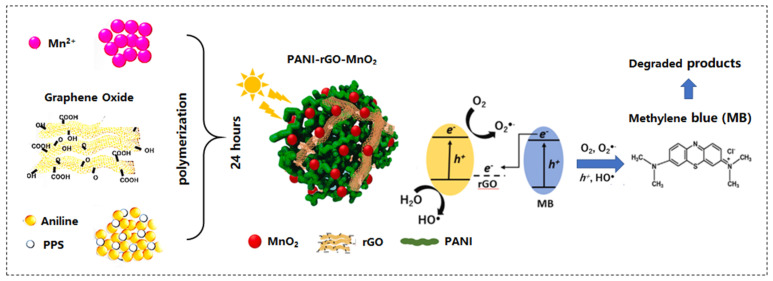
Application schematic illustration of the ternary PANI-rGO-MnO_2_ composite for photocatalytic degradation of organic dye MB under sunlight irradiation. Figures reprinted with permission from ref. [250]. Copyright 2021; Elsevier Ltd.

**Table 1 nanomaterials-13-00541-t001:** Summary of preparation methods, products, and morphological characteristics of synthesizing MnO_2_-carbon materials.

MnO_2_	Carbon Material	Synthesis Method	Composite Product	Morphology	Ref.
ultrafine MnO_2_ nanowires	CC	hydrothermal	MnO_2_@CC	Weedy 1D ultrafine MnO_2_ nanowire interconnection network covered on the surface of CC.	[74]
MnO_2_	g-C_3_N_4_	In situ redox deposition	MnO_2_/g-C_3_N_4_	flower-like MnO_2_ nanosheets deposited on g-C_3_N_4_, resulting in surface roughness.	[125]
MnO_2_	3D Graphene Networks	Electrochemical deposition	3D Graphene/MnO_2_	MnO_2_ nanoporous structures were uniformly coated on a 3D graphene network skeleton.	[146]
α-MnO_2_	HMCNTs	Co-precipitating	MnO_2_/HMCNTs	MnO_2_ was deposited on the surface of CNTs and provided active sites.	[154]
MnO_2_	g-C_3_N_4_	Sonochemical	g-C_3_N_4_/MnO_2_	Different sizes of materials were obtained by ultrasound with different amplitudes.	[169]
MnO_2_ Polyhedron Precursors	Bulk-g-C_3_N_4_ nanosheets	Calcination	3D/2D MnO_2_/g-C_3_N_4_ Nanocomposite	MnO_2_ was wrapped by the g-C_3_N_4_ layers.	[171]
MnO_2_ Nanorods	Mn-modified alkalinized g-C_3_N_4_	Impregnation	Z-scheme MnO_2_/Mn-modified alkalinized g-C_3_N_4_ heterojunction	In the process of Mn modifying alkalinized g-C_3_N_4_, slender rod-shaped MnO_2_ was formed.	[172]
layered MnO_X_	GO	hydrothermal	GO/MnO_X_ composites	nanosheets	[75]
α-MnO_2_ nanorods	MWCNTs	direct pyrolysis	MWCNTs/MnO_2_ nanocomposite	MnO_2_ nanorods are uniformly attached to the surface of MWCNTs.	[173]

**Table 2 nanomaterials-13-00541-t002:** Study on MnO_2_-carbon materials for photocatalytic degradation of phenolic compounds in aqueous solution.

Photocatalyst	Target Pollutant	Light Source	Photocatalyst Amount	Initial Concentration	Activity	Ref.
Titanium dioxide-manganese oxide/multi-walled CNT (TiO_2_-MnO_2_/ MWCNT)	phenol	UV light 150 W fluorescent lamp	90 mg	300 mL 100 mg/L	40 min 100%	[208]
CQDs decorated MnO_2_ nanorods (MnO_2_@CQDs)	phenol	visible light	/	100 mg/L	50 min 90%	[207]
MnO_2_/g-C_3_N_4_ (MG3)	phenol	visible light	50 mg	100 mL 5 mg/L	100 min 98%	[209]
2D g-C_3_N_4_/MnO_2_ heterojunctions (2D g-C_3_N_4_/MnO_2_)	phenol	visible light 300 W Xenon lamp	50 mg	50 mL 50 mg/L	180 min 73.6%	[157]
2D/1D protonated g-C_3_N_4_/α-MnO_2_ (CNM)	phenol	visible light 300 W Xe arc lamp	40 mg	80 mL 10 mg/L	120 min 93.8%	[67]
g-C_3_N_4_/MnO_2_/Pt	Phenol; Bisphenol A	Solar source 300 W Xenon lamp	50 mg 20 mg PMS	100 mL 20 mg/L	30 min 20%→57%; 13%→97%	[210]
Dye-loaded MnO_2_ and chlorine-intercalated g-C_3_N_4_ (MO/CN-Cl)	Phenol; 2,4-dichlorophenol	visible light 150 W Xe lamp	200 mg	50 mL 20 mg/L	1 h 47%; 1 h 60%	[211]
Graphene oxide/MnO_2_ nanocomposite (rGO/MnO_2_)	2-naphthols	visible light 20 W LED	100 mg	144 mg	12 h 97.2%	[193]
3 wt% MnO_2_ modified exfoliated porous g-C_3_N_4_ nanosheet (GM3)	aromatic alcohols	visible light 150 W xenon lamp	/	20 mL 100 mg/L	80 min 78%	[212]

**Table 3 nanomaterials-13-00541-t003:** Study on MnO_2_-carbon materials for photocatalytic degradation of antibiotic in aqueous solution.

Photocatalyst	Target Pollutant	Light Source	Photocatalyst Amount	Initial Concentration	Activity	Ref.
Porous Z-scheme MnO_2_/Mn-modified alkalinized g-C_3_N_4_ heterojunction (MnO_2_/CNK-OH-Mn_15%_)	tetracycline	visible light 300 W Xe lamp	50 mg	100 mL 10 mg/L	120 min 96.7%	[172]
Carbon nanosheet/MnO_2_/BiOCl (Cs/Mn/Bi-1/1)	tetracycline hydrochloride	UV light 300 W mercury lamp	20 mg	100 mL 20 mg/L	30 min 80%	[225]
g-C_3_N_4_/diatomite/MnO_2_	tetracycline hydrochloride	visible light	30 mg	100 mL 50 mg/L	60 min 87%	[226]
g-C_3_N_4_/MnO_2_/GO (CMG-10)	tetracycline hydrochloride	visible light 300 W xenon lamp	50 mg	100 mL 10 mg/L	60 min 91.4%	[227]
g-C_3_N_4_-MnO_2_ (CMn_2_)	tetracycline hydrochloride	visible light LED	30 mg	75 mL 20 mg/L	135 min 92.47%	[169]
Pumice-loaded rGO@MnO_2_ PS@rGO@MnO_2_	ciprofloxacin	sunlight 300 W xenon lamp	300 mg	30 mL 5 mg/L	6 h 80%	[228]
g-C_3_N_4_/MnO_2_/Pt	sulfadiazine	Solar source 300 W Xenon lamp	50 mg 20 mg PMS	100 mL 20 mg/L	30 min 11%→68%	[210]

**Table 4 nanomaterials-13-00541-t004:** Study on MnO_2_-carbon materials for photocatalytic degradation of organic dye in aqueous solution.

Photocatalyst	Target Pollutant	Light Source	Photocatalyst Amount	Initial Concentration	Activity	Ref.
MnO_2_/CNT	MB	visible light solar radiation	20 mg	50 mL 20 mg/L	75 min 70%	[239]
Cu-doped MnO_2_/r-GO	MB	visible light 200 W tungsten bulb	20 mg	50 mL 5 mg/L	90 min 86.69%	[240]
PANI-rGO-MnO_2_	MB	visible light 150 W halogen bulb with Halogen cold light source	10 mg	5 mg/L	120 min 91%	[250]
MnO_2_/BC	MB	27 °C sunlight 45 °C	10 mg	10 mL 10 mg/L	120 min 85% 97%	[241]
α-MnO_2_ nanowire/activated carbon hollow fibers (MnO_2_@ACHF)	MB	visible light	20 mg	33 mg/L	240 min 99.8%	[38]
poly(3, 4-ethylenedioxythiophene)/GO/MnO2 (PEDOT/GO/MnO_2_)	MB	UV light sunlight	20 mg	50 mL	7 h 97.1% 7 h 98.9%	[242]
graphene nano sheets/CNT/MnO_2_ (GNS/CNT/MnO_2_)	MB MG	visible light 400 W metal Philips lamp	60 mg	250 mL 60 mg/L	60 min 71% 60 min 89%	[243]
GO@Fe_3_O_4_-MnO_2_	MG tartrazine	sunlight	10 mg	50 mL 10 mg/L	70 min 99.9% 80 min 98%	[244]
Carbon nanosheet/MnO_2_/BiOCl (Cs/Mn/Bi-1/1)	RhB MB	UV light 300 W mercury lamp	10 mg	100 mL 10 mg/L	25 min 97% 40 min 98%	[225]
g-C_3_N_4_/diatomite/MnO_2_	RhB	visible light	30 mg	100 mL 10 mg/L	50 min 94%	[245]
2D/1D protonated g-C_3_N_4_/α-MnO_2_ (CNM)	RhB	visible light 300 W Xe arc lamp	40 mg	80 mL 10 mg/L	60 min 98.8%	[67]
2D g-C_3_N_4_/MnO_2_	RhB	visible light 300 W Xenon lamp	50 mg	50 mL 10 mg/L	60 min 91.3%	[157]
MnO_2_@GO (MG _0.4_)	RhB	visible light 500 W xenon–mercury lamp	40 mg	50 mL 20 mg/L	65 min 93.86%	[246]
g-C_3_N_4_/MnO_2_ (GCN/MnO_2_)	RhB	sunlight	4 mg	20 mL 9.6 mg/L	90 min 100%	[247]
Boron-doped carbon nitrides/MnO_2_ (BCN/MnO_2_)	RhB	visible light	25 mg	50 mL 10 mg/L	180 min 61.1%	[248]
g-C_3_N_4_/MnO_2_/Pt	RhB MO	Solar source 300 W Xenon lamp	50 mg 20 mg PMS	100 mL 20 mg/L	30 min 99% 30 min 97%	[210]
nitrogen-doped grapheme/MnO_2_ NG-MnO_2_	MO	visible light	5 mg	5 mL 20 mg/L	70 min 95%	[77]
MnO_2_/g-C_3_N_4_ (MG3)	MO	visible light	50 mg	100 mL 5 mg/L	100 min 92%	[251]
Fe_3_O_4_/C/MnO_2_/C_3_N_4_	MO	400 W metal halide lamp	20 mg	20 mL 10 mg/L	140 min 94.11%	[249]

## Data Availability

Where no new data were created.

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
