# Peer review of "Preparation of MnO2-Carbon Materials and Their Applications in Photocatalytic Water Treatment"

_nanomaterials, 2023, doi:10.3390/nano13030541_

Round 1

Reviewer 1 Report

He authors have collected manganese dioxide and different kinds of carbon material composites for the application in the treatment of water pollutants. The authors have also provided their perspective on the challenges and future research directions of MnO2-carbon materials in the field of environmental applications. This review can inspire more material design ideas of MnO2-carbon materials for environmental applications. Overall, this is a well-written and well-organized review paper. Therefore, I would like to recommend this review to publish in Nanomaterials. I only have one suggestion for the authors. For the introduction “Photocatalytic materials are the core of photocatalytic technology”, more references could be cited to broaden the introduction.

https://doi.org/10.1021/acsmaterialslett.2c00752

https://doi.org/10.1002/anie.202211142

https://doi.org/10.1039/C8TA11471D

Author Response

Dear Editors and Reviewers: 

Thank you for your letter and for the reviewers’ comments concerning our manuscript entitled “Preparation of MnO2-Carbon Materials and Their Applications in Photocatalytic Water Treatment”. Those comments are all valuable and very helpful for revising and improving our paper, as well as the important guiding significance to our researches. We have studied comments carefully and have made correction which we hope meet with approval. The main corrections in the paper and the responds to the reviewer’s comments are as follows: 

Point 1:For the introduction “Photocatalytic materials are the core of photocatalytic technology”, more references could be cited to broaden the introduction.

https://doi.org/10.1021/acsmaterialslett.2c00752

https://doi.org/10.1002/anie.202211142

https://doi.org/10.1039/C8TA11471D

 Response1Thank you very much for your comments. For the introduction “Photocatalytic materials are the core of photocatalytic technology”, we have added new references to further substantiate this point and have read and added above three references. (Page4; Line 12)

Reviewer 2 Report

The interesting work could be published on this scientific issue after thorough structural and linguistic revision.

In current study, photocatalytic technology has shown to gain attention using MnO2. Photocatalytic technology applying MnO2 can remove antibiotics, heavymetals, dyes, having savings without energy and carbon demand-  could combination with heterotrophic nitrification-aerobic denitrifying bacteria can simultaneously accomplish enhanced nitrification and denitrification while using currently synthesized MnO2 carbon materials? What pollutants can be removed specifically by used catalytic materials need to be defined in abstract and elsewhere? Do bacterial removal occurrs also or materials are biofriendly?

Following questions have arisen:

1. The title should say something about novel result of the research and show the innovative result. This application is not definitely increasing the efficiency of the system alone when not tested for different pollutant classes. Limit the number of figures and tables, give only the most important one’s results. Error bars are mandatory in figures (Fig. 3). Color legend in Fig. 2, 3 with magnification bar is not visible well. Graphical abstract is missing. Normally, chemical results are shown first, then showing characterization techniques. Figures should be made on Your own not copied from several works.

Table 1 showing concentrations could be done in mg/L, not in mixed form of ppm and ml and mmol. Need to check whether using 0.3 g photocatalyst was right for treatment of 100 ppm phenoles, seems unrealistic.

Adsoprtion capacities g/m2 need to be given.

Pollutants removal rates per m3 reactor or m2 catalytc material could be given in addition to removal %.

2. “manganese dioxide (MnO2) is the most promising photocatalytic candidate material due to its narrow band gap (0.9-1.2eV), low cost, non-toxic properties, ease of synthesis, large surface area, rich structures and morphologies, outstanding adsorption and oxidation capacity “The interpretation of what narrow band gap means compared to other semiconductors needs clarifying as well as other parameters, how much better these qualities are for comparison with other semi conductors? Potential applied concentrations for treatment are needed to be shown as well in different removal%.

The choice of alternative fillers and chosen applicable pollutants concentrations needs to be added and conditions need clarifying. Are there other fillers potentially be used that can work better, in any microbial technology without disturbances? How did removal rates differ? What were the measures in effluent for N, COD, BOD? The average and stage N, COD, BOD removal rates per m2 catalytic material/m3 reactor need to be added.

carbon materials and MnO2 – there are mistakes in super and subscripts used.

First we summarizes – should be “we summarize”

Surface area and adsorption capacities need to be shown numerically.

Hypothesis and aims are missing from current MS, need to add these. Novelty aspects as well. Introduction section does not state Your own results

In my opinion, main point of this study was somehow missed providing extensive discussion on too many details on bacterial data of characteristics, but not about main aims pointed.

 Language and structuring of the work should be substantially revised.

There should be some specific conclusions and the main point of the work better introduced.

Without concrete values and more sophisticated statistical techniques the text of the abstract and other sections remains vague. Word order of the sentences needs revision.

Abbreviations in the manuscript body, at the first occurrence, should be in abbreviated form plus full definition; then they should be given only in abbreviated forms throughout the manuscript.

Check and add recent reports on adsorption and removal of dyes, antibiotics: DOI: https://doi.org/10.3390/w13111522, https://doi.org/10.3390/w13141969, https://doi.org/10.3390/w13152136

References authors are in capital letters in current manuscript, which is not commonly used in journals, revise. There are some that are not bold, unify and revise other referencing mistakes as per journal requirements given. References are not suppose to be with different colors also in text. Refs should be at the end of sentence normally.

Author Response

Dear Editors and Reviewers: 

Thank you for your letter and for the reviewers’ comments concerning our manuscript entitled “Preparation of MnO2-Carbon Materials and Their Applications in Photocatalytic Water Treatment”. Those comments are all valuable and very helpful for revising and improving our paper, as well as the important guiding significance to our researches. We have studied comments carefully and have made correction which we hope meet with approval. Please see the attachment.

Reviewer 3 Report

Preparation of MnO2-Carbon Materials and Their Applications in Photocatalytic Water Treatment

This manuscript studies the use of MnO2-carbon materials in the field of environmental applications.

The topic is interesting but there are some aspects that the authors should improve. The manuscript could be published after major revision.

Keywords: I suggest to add also MnO2-carbon materials

Abstract: Line 21-22 should be changed specify with more details the content of the review. For example “This review covers recent progress on MnO2-carbon materials for photocatalytic water treatment. The details of MnO2 - carbon materials are summarized followed by an overview of different methods adopted for the preparation of composites. Special attention is paid to their applications in removal of phenolic compound, antibiotic dyes and heavy metal ions removal”.

Introduction:

1.      Please change the word “review” instead “paper”

2.      The introduction should be revised in order to underline the best crystalline structure of MnO2 for photocatalytic applications.

3.      Why for all the semiconductor TiO2, ZnO,Bi2O3,Fe2O3,SnO2, MnO2 is the best one to combine with carbon material? Please underline this aspect in the revised manuscript. The content of this manuscript needs to be further enriched.

 4.      Line 78 “MnO2 “instead “MnO2”

Preparation methods of MnO2-Carbon composites:

  1. The authors could be add a table reported all sample synthesis methods and main physiochemical characterization for example crystalline phase of Mn2O, specific surface area and so on .
  2. The composite materials seem work also under visible light and direct solar light, please underline also this aspect adding a consideration about the band-gap values.
  3. The photoelectrochemical performance and steady-state fluorescence are crucial to explaining the carrier transport and recombination of photocatalytic materials, please give more details also about this ascpect
  4. It is important moreover adding a specify paragraph related to the formulation of  ternary composites for example PANI-rGO-MnO2

Applications of Photocatalytic Technology in water treatment

1. Some details about the stability of composites systems should be provided

2. What is the function of third element (for example PANI) during photocatalytic process?

Conclusions and outlook

In the section of concluding remarks, the author should discuss more on the present challenges and prospects of the system Mn2O-carbon and not in generally the present challenges and prospects of photocatalytic process.

Author Response

(The authors gave the same response as above.)

Round 2

Reviewer 3 Report

The authors carefully response for all questions. So, I think that now the paper should be published in the nanomaterial journal